# MiNet: Weakly-Supervised Camouflaged Object Detection through Mutual Interaction between Region and Edge Cues

Yuzhen Niu
Fuzhou University
Fuzhou, China
yuzhenniu@gmail.com

Lifen Yang
Fuzhou University
Fuzhou, China
yanglifena@gmail.com

Rui Xu*
Fuzhou University
Fuzhou, China
xurui.ryan.chn@gmail.com

Yuezhou Li
Fuzhou University
Fuzhou, China
liyuezhou.cm@gmail.com

Yuzhong Chen
Fuzhou University
Fuzhou, China
yzchen@fzu.edu.cn

## Abstract

Existing weakly-supervised camouflaged object detection (WSCOD) methods have much difficulty in detecting accurate object boundaries due to insufficient and imprecise boundary supervision in scribble annotations. Drawing inspiration from human perception that discerns camouflaged objects by incorporating both object region and boundary information, we propose a novel Mutual Interaction Network (MiNet) for scribble-based WSCOD to alleviate the detection difficulty caused by insufficient scribbles. The proposed MiNet facilitates mutual reinforcement between region and edge cues, thereby integrating more robust priors to enhance detection accuracy. In this paper, we first construct an edge cue refinement net, featuring a core region-aware guidance module (RGM) aimed at leveraging the extracted region feature as a prior to generate the discriminative edge map. By considering both object semantic and positional relationships between edge feature and region feature, RGM highlights the areas associated with the object in the edge feature. Subsequently, to tackle the inherent similarity between camouflaged objects and the surroundings, we devise a region-boundary refinement net. This net incorporates a core edge-aware guidance module (EGM), which uses the enhanced edge map from the edge cue refinement net as guidance to refine the object boundaries in an iterative and multi-level manner. Experiments on CAMO, CHAMELEON, COD10K, and NC4K datasets demonstrate that the proposed MiNet outperforms the state-of-the-art methods.

## CCS Concepts

• **Computing methodologies → Scene understanding**.

## Keywords

Camouflaged object detection; Region guidance; Edge guidance

---

*Corresponding author

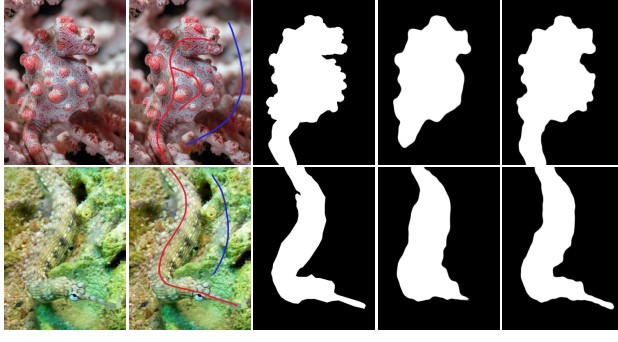

**Figure 1: Compared with CRNet [13], our method gives more accurate localization of the camouflaged object boundaries.**

Input Scribble GT Pixel-wise GT CRNet [13] Ours

## ACM Reference Format:

Yuzhen Niu, Lifen Yang, Rui Xu, Yuezhou Li, and Yuzhong Chen. 2024. MiNet: Weakly-Supervised Camouflaged Object Detection through Mutual Interaction between Region and Edge Cues. In *Proceedings of the 32nd ACM International Conference on Multimedia (MM '24), October 28-November 1, 2024, Melbourne, VIC, Australia.* ACM, New York, NY, USA, 10 pages. https://doi.org/10.1145/3664647.3680891

## 1 Introduction

Camouflaged object detection (COD) aims to identify and separate objects hidden within their surroundings. It has attracted significant attention due to the potential applications in various fields such as medical diagnosis [7, 15], species protection [26], and crop pest detection [22]. Unlike conventional object detection [6, 11], COD faces more rigorous challenges. The high intrinsic similarity between the camouflaged objects and their surroundings demands that COD can discern object internal information based on fine-grained details. Furthermore, as a pixel-level classification task, COD requires more precise boundary detection results.

In recent years, there has been a growing interest in leveraging data-driven deep learning techniques for COD. Fully-supervised methods, which rely on pixel-wise annotations, have achieved significant advances. Nevertheless, there are still potential obstacles that hinder the widespread application of these methods in COD.

Firstly, the manual pixel-wise annotation for large-scale datasets is both time-consuming and labor-intensive. In addition, the conventional pixel-wise annotation methods that treat each pixel within the object region equally may not adequately capture the essential structural characteristics of the object [13]. To overcome these obstacles, sparse annotation methods have emerged to streamline dataset annotation and mitigate overfitting. Consequently, exploring weakly-supervised camouflaged object detection (WSCOD) methods that leverage sparse annotations as supervision could be a promising avenue.

However, the restricted annotations information available for WSCOD also significantly impedes the detection performance. He *et al.* [13] propose a pioneering scribble annotation method by sketching the main structure of the foreground and background regions, providing greater flexibility and significantly reducing the time and labor costs of dataset annotation. However, scribble annotations are difficult to provide sufficient reference information to infer the precise boundaries of camouflaged objects. To tackle this issue, He *et al.* [13] further introduce a consistency loss to attain reliable consistency cross different images and inside a single prediction map. Nevertheless, the loss is calculated based on image features that are not descriminative enough, potentially resulting in inaccurate localization of object boundaries. As shown in Fig. 1, the prediction results of CRNet [13] exhibit a significant deficiency of the boundary structures for the camouflaged objects.

To tackle the mentioned issue, another strategy to improve object boundary accuracy is to incorporate the edge prior as a supplementary aid [42, 44, 50]. For instance, the scribble-based weakly-supervised salient object detection method [44] has successfully aided the learning of object boundaries by introducing the edge detection task. However, in contrast to salient object images, most camouflaged object images have cluttered backgrounds. As demonstrated in Fig. 2, the edge map obtained from the edge detection network for COD image may contain a substantial amount of non-object noise, which may negatively mislead COD task. Therefore, it is essential to provide the discriminative edge prior for scribble-based COD. Moreover, earlier biological research [10] has offered fresh perspectives on WSCOD. The study reveals that human perception of camouflaged objects entails a sequential process: first identifying the rough region of the camouflaged objects, then focusing attention to delineate their boundaries, and ultimately integrating these boundaries with the object regions to effectively separate the camouflaged objects from the backgrounds.

In this paper, inspired by human perception, we propose a novel Mutual Interaction Network (MiNet) for scribbled-based WSCOD. Specifically, drawing inspiration from human cognitive processes, our MiNet is intricately designed to utilize mutual reinforcement between region and edge cues, generating distinctive cues to improve the boundary prediction results of camouflaged objects. To achieve this goal, we first construct an *edge cue refinement net*. Within this net, a coarse edge detection block (EDB) first aggregates multi-level features from the backbone to obtain coarse edge feature. Subsequently, a region-aware guidance module (RGM) highlights the regions associated with the object within the edge feature. RGM takes into account both the object semantic and positional relationships between edge feature and region feature, resulting in a discriminative edge map. To address the issue of inaccurate

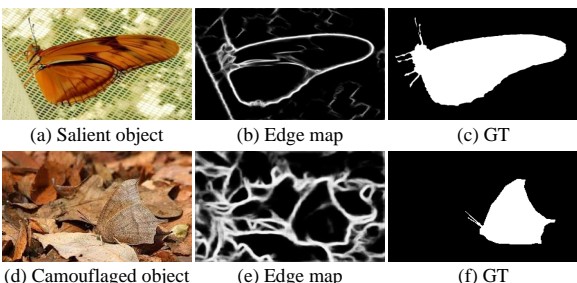

|  |  |  |
|---|---|---|
| (a) Salient object | (b) Edge map | (c) GT |
| (d) Camouflaged object | (e) Edge map | (f) GT |

**Figure 2: Examples of extracted edge maps with edge detector [23] from salient and camouflaged object images.**

boundary localization caused by the high similarity between the camouflaged objects and their surroundings, we further devise a *region-boundary refinement net*, which uses an edge-aware guidance module (EGM) at each network level. EGM utilizes the edge map as prior to guide the network to focus more on object boundaries, thereby generating more discriminative region feature. Moreover, we employ an iterative refinement manner by propagating the discriminative region features to the next level of the network, where they are fused with backbone features and enhanced through the introduced residual inception block [19]. After iterative refinement, our method achieves more accurate object boundary localization.

In summary, our contributions are summarized as follows:

- We propose a biology inspired WSCOD method, which enhances the accuracy of object boundary prediction through mutual interaction between region and edge cues.
- We devise the region-aware guidance module embedded within the edge cue refinement net, deliberately leveraging the semantic and positional information of region cue to enhance the discriminability of the edge map.
- We develop the edge-aware guidance module integrated into the region-boundary refinement net, leveraging the discriminative edge map as guidance for multi-level and iterative refinement of object boundaries.
- Experimental results on four benchmark datasets demonstrate the superior performance of MiNet.

## 2 Related Work

### 2.1 Camouflaged Object Detection

Traditional COD methods attempt to extract the hand-crafted features, such as color [14] and texture [8], to describe the camouflaged objects. However, these methods often fail in complex scenes where the camouflaged objects are highly similar with their surroundings. Due to the advances of deep learning and the availability of benchmark datasets [5, 18, 24], more and more deep learning based COD methods have emerged.

Specifically, some methods [5, 16, 29] employ biomimetic approaches to detect camouflaged objects by imitating human or animal behaviors. For instance, Fan *et al.* [5] design the SINet inspired by the two-stage process of predators hunting, which includes a search module and an identification module. In addition, several methods leverage the multi-task learning strategy. For example, Zhai *et al.* [42] corporate the COD task and the edge detection

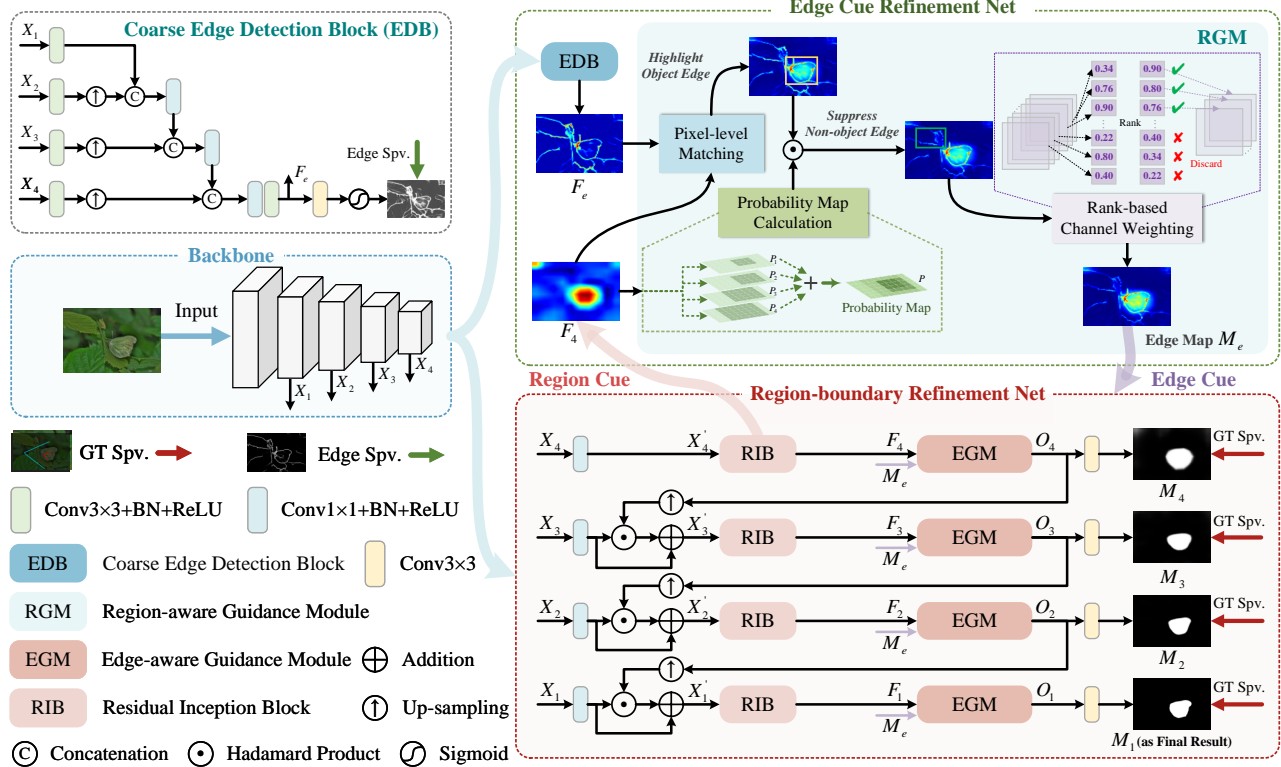

**Figure 3: Overall framework of the MiNet. Specifically, the MiNet consists of the backbone for multi-level feature extraction, the edge cue refinement net for obtaining discriminative edge map via EDB and RGM, and the region-boundary refinement net for refining the object boundaries via EGMs in an iterative and multi-level manner.**

task, introducing the graph-based model to iteratively reason their relationship. Considering that relying solely on RGB domain information makes it difficult to accurately locate camouflaged objects in challenging scenes, some works [33, 47] improve the performance of COD by introducing additional information as an aid. For instance, Zhong *et al.* [47] leverage the frequency domain features to assist in distinguishing the subtle differences between the camouflaged objects and the background.

The above fully-supervised learning-based methods have achieved certain advancements. However, these methods heavily rely on pixel-wise ground truth and may overlook the primary structure of the object. Therefore, exploring weakly-supervised camouflaged object detection methods based on sparse annotations is a worthwhile problem to investigate.

## 2.2 Weakly-supervised Learning with Scribbles

Weakly-supervised learning with scribbles has been studied in some other tasks, such as semantic segmentation [20, 28, 32] and salient object detection (SOD) [41, 44]. Lin *et al.* [20] propagate the scribble information to other unknown regions based on super-pixels and graph-cuts methods to obtain the fully labeled pseudo label. Pan *et al.* [28] address the inconsistency problem by reducing the uncertainty of neural representations and introducing self-supervision strategy. Zhang *et al.* [44] introduce edge detection network to assist in weakly-supervised SOD based on scribble. Yu *et al.* [41]

propose a local coherence loss that propagates scribble throughout the entire image based on color information.

However, the above scribble-based learning methods, specifically tailored for semantic segmentation and SOD, are generally not suitable for the COD task. Firstly, semantic segmentation as a pixel-wise classification task, typically involves hundreds or thousands of categories, whereas COD is a binary pixel classification task. Secondly, due to the inherent similarity between camouflaged objects and their surroundings, directly applying semantic segmentation or SOD methods to COD usually achieves inferior performance. To address the mentioned issues, based on the first scribbled-based dataset (S-COD) for WSCOD, He *et al.* [13] present a scribble-based framework, which extends scribble to wider camouflaged areas by utilizing low-level contrasts, while also leveraging semantic relationships to determine the regions of camouflaged objects. However, due to the sparsity of scribble annotations and the lack of explicit guidance from edge cue, this method has difficulty in accurately locating the boundaries of camouflaged objects.

Taking into consideration the challenges mentioned above and inspired by human behavior in identifying camouflaged objects, we propose to mine the relationship between region and edge cues. Specifically, our proposed method leverages the mutual interaction between region and edge cues to achieve more discriminative features and more accurate detection results.

# 3 Proposed method

## 3.1 Network Overview

As shown in Fig. 3, the proposed MiNet comprises three primary sub-networks: a backbone net for multi-level feature extraction, an edge cue refinement net to enhance the coarse edge feature, and a region-boundary refinement net to improve boundary accuracy of the detected camouflaged objects. Leveraging the capability of ResNet-50 [12], the backbone net extracts multi-level features $X_i$ ($i \in \{1, 2, 3, 4\}$), thereby furnishing abundant contextual information crucial for identifying camouflaged objects.

In the edge cue refinement net, the EDB is used to integrate the extracted multi-level features and generate the coarse edge feature $F_e$. Furthermore, the RGM incorporates the region cue (i.e., the region feature $F_4$) from the region-boundary refinement net to suppress non-object noise within the edge feature $F_e$, thus obtaining a more discriminative edge map $M_e$.

In the region-boundary refinement net, the RIB and the EGM are embedded in the each level of the network. The RIB first enhances the feature $X_i'$ to produce region feature $F_i$. Subsequently, the EGM utilizes the edge cue (i.e., the edge map $M_e$) as guidance to enhance $F_i$, resulting in the discriminative region feature $O_i$. Furthermore, by propagating each $O_i$ ($i \in \{2, 3, 4\}$) to the $i$-1-$th$ level of the network, object boundaries can be refined in an iterative and multi-level manner. Finally, based on the region feature $O_i$, a $3 \times 3$ convolutional layer is employed to derive the result $M_i$ ($M_1$ is the final result).

The pink and purple arrows between the edge cue refinement net and the region-boundary refinement net in Fig. 3 illustrate the mutual interaction between region and edge cues. Specifically, the region cue help to achieve more discriminative edge map, which in turn help to locate the object boundaries accurately.

## 3.2 Edge Cue Refinement Net

In the edge cue refinement net, as shown in Fig. 3, EDB is first used to generate the coarse edge feature $F_e$. As the presence of non-object edge noise in $F_e$ may mislead the COD task, it's necessary to refine the edge feature to obtain discriminative edge prior. Specifically, the RGM suppresses the non-object noise in $F_e$ with the guidance of region cue (i.e., the region feature $F_4$) from the region-boundary refinement net, resulting in an enhanced edge map $M_e$. And $M_e$ will be fed as edge cue into the region-boundary refinement net to make the prediction of object boundaries more accurate.

**Coarse Edge Detection Block (EDB).** EDB extracts the coarse edge feature and further estimates the coarse edge map. As shown in the top left corner of Fig. 3, each of the backbone feature $X_i$ ($i \in \{1, 2, 3, 4\}$) is processed by a $3 \times 3$ convolutional block, and then up-sampled to the same size. These features, which incorporate rich edge details and high-level semantic information, are then progressively aggregated through concatenation operations and $1 \times 1$ convolutional blocks. Subsequently, a $3 \times 3$ convolutional block is applied to derive the coarse edge feature $F_e$. In addition, a $3 \times 3$ convolutional layer and a Sigmoid function are performed on $F_e$ to generate the coarse edge map $e$. Here, the coarse edge feature $F_e$ is taken as the input of RGM, and the coarse edge map $e$ is supervised by the edge ground truth $E$ of the camouflaged image, which is obtained by the edge detector [23]. Examples of extracted edge maps $E$ are illustrated in Fig. 2.

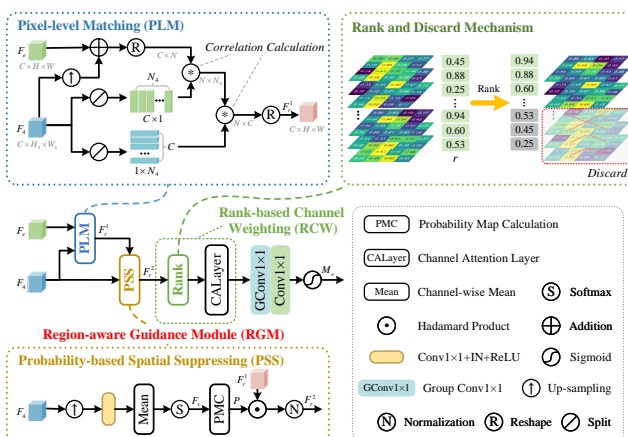

**Figure 4: Structure of the region-aware guidance module.**

**Region-aware Guidance Module (RGM).** RGM highlights the object edge and suppresses the non-object noise by taking into account the object semantic and positional information relationships between edge feature and region feature. By employing a combination of pixel-level matching operation, probability-based spatial suppressing operation, and rank-based channel weighting operation, the module achieves the enhanced edge map $M_e$.

Specifically, as illustrated in Fig. 4, a pixel-level matching operation is first employed to highlight areas within the edge feature exhibiting high semantic correlation with the region feature (see the yellow anchor box in the feature heatmap visualization in Fig. 3). However, the region feature primarily emphasizes holistic and semantic information, whereas the edge feature concentrates more on object structures. This results in notable disparities in their numerical distributions and feature content. To tackle this issue, we integrate the edge feature with the region feature, thereby mitigating the differences between them while retaining their distinct characteristics. In detail, after up-sampling the region feature $F_4 \in \mathbb{R}^{C \times H_4 \times W_4}$ to match the size of $F_e \in \mathbb{R}^{C \times H \times W}$, these two features are then added to obtain edge feature $F_e' \in \mathbb{R}^{C \times H \times W}$. Considering the disruptive effect of cluttered background noise on correlation calculations based on non-local spatial attention [1], we split the reshaped region feature along both channel and spatial dimensions. The correlations are then calculated in a spatial-wise and channel-wise manner sequentially. In particular, the decomposed region feature contains $N_4 = H_4 W_4$ spatial feature vectors of size $C \times 1$ and $C$ channel feature vectors of size $1 \times N_4$, denoted as $F_{4s} = \{F_{4s}^1, F_{4s}^2, \cdots, F_{4s}^{N_4}\}$ and $F_{4c} = \{F_{4c}^1, F_{4c}^2, \cdots, F_{4c}^C\}$, respectively. Here, $F_{4s}^j \in \mathbb{R}^{C \times 1}$, $j \in \{1, 2, ..., N_4\}$, $F_{4c}^k \in \mathbb{R}^{1 \times N_4}$, $k \in \{1, 2, ..., C\}$. Then, $F_e'$ is reshaped into $F_e' \in \mathbb{R}^{C \times N}$, where $N = HW$. Subsequently, the correlation between edge feature $F_e'$ as well as spatial feature vectors $F_{4s}$ and channel feature vectors $F_{4c}$ can be calculated in a sequentially manner, which can be expressed as follows:

$$F_r^1[p, q] = \sum_{o,p} F_e'[p, q] \cdot F_{4s}[o, p] \cdot F_{4c}[p, o], \quad (1)$$

where $p$ denotes the channel index, $q$ and $o$ denote the spatial index, $p \in [1, C]$, $q \in [1, N]$, and $o \in [1, N_4]$, respectively. $F_r^1 \in \mathbb{R}^{N \times C}$ denotes the matching result. Then $F_r^1$ is reshaped to $F_r^1 \in \mathbb{R}^{C \times H \times W}$.

Furthermore, in some complex scenarios, the pixel-level matching operation may lead to undesired background enhancement. To address this, we design the probability-based spatial suppressing operation to incorporate positional information for helping further suppress non-object noise, while retaining the uncertainty of the object localization provided by the region feature $F_4$. This operation includes the probability map calculation and spatial suppressing via Hadamard product. Specifically, considering the uncertainty of the object localization, we assign different levels of probability value to each pixel. In detail, $F_4$ is up-sampled to match the size of $F_r^1$, then processed by a $1 \times 1$ convolutional block. After that, a channel-wise mean operation and a Softmax function are applied to yield the feature $F_s \in \mathbb{R}^{1 \times H \times W}$. For the top $s_i$ ( $i \in \{1, 2, 3, 4\}$, $s_i \in \{\frac{1}{2}HW, \frac{2}{3}HW, \frac{3}{4}HW, \frac{4}{5}HW\}$ ) pixels with the highest values in $F_s$, a probability weight of 1 is assigned, while all other pixels receive a weight of 0.5, resulting in a probability map $P_i \in \mathbb{R}^{1 \times H \times W}$. Then different $P_i$ are summed to produce a combined probability map $P \in \mathbb{R}^{1 \times H \times W}$. Subsequently, a spatial suppressing process is performed by Hadamard product between $F_r^1$ and $P$, followed by a normalization operation to produce edge feature $F_r^2$.

To acquire a more discriminative edge map, we further present a lightweight yet effective rank-based channel weighting operation. This operation leverages a parameter-free rank-based mechanism to discard some less informative channels. The rank and discard mechanism is illustrated in the top right corner of Fig. 4. Through a global max pooling, ranking score for each channel in $F_r^2$ are derived, yielding a ranking score vector $r \in \mathbb{R}^{1 \times C}$. Then, the channels are ranked based on the values in $r$, and the top half of channels with higher ranking scores are retained while the remaining channels are discarded. Moreover, with a channel attention layer, the retained channels are re-weighted to highlight the object information. Finally, two convolutional layers followed by a Sigmoid function are applied to generate the discriminative edge map $M_e$.

## 3.3 Region-boundary Refinement Net

Due to the inherent similarity between the camouflaged objects and their surroundings, accurately distinguishing their boundaries poses a certain level of difficulty. To address this issue, within the region-boundary refinement net, the edge map $M_e$ engages in multi-level interactions with region features, consequently iteratively refining the object boundaries.

In this sub-network, RIB is first used to explore object information from multi-receptive fields by using convolutional layers of different kernel sizes, detailed structure of RIB can be found in [19]. More specifically, for the deepest-level, RIB takes the region feature $X_4'$ (which is obtained by projecting $X_4$ via a convolutional block) as input and produces region feature $F_4$. Subsequently, the EGM enhances $F_4$ to obtain the discriminative region feature $O_4$. For the other three levels, $O_{i+1}$ ( $i \in \{1, 2, 3\}$) from the higher level is propagated to the $i$-th level of the network, fused with the feature from the backbone, and then the combined feature is fed into the RIB as follows:

$$
\begin{cases}
F_i = RIB\left(\mathcal{C}\left(X_i\right)\right), i = 4 \\
F_i = RIB\left(\mathcal{C}\left(X_i\right) \oplus \left(\mathcal{C}\left(X_i\right) \odot \mathcal{U}\left(O_{i+1}\right)\right)\right), i \in \{1, 2, 3\}, \\
O_i = EGM\left(F_i, M_e\right), i \in \{1, 2, 3, 4\}
\end{cases}
\tag{2}
$$

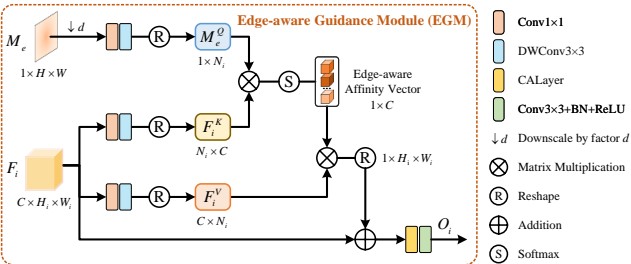

**Figure 5: Structure of the edge-aware guidance module.**

where $\mathcal{C}\left(\cdot\right)$ represents the $1 \times 1$ convolutional block, $\mathcal{U}\left(\cdot\right)$ represents the up-sampling operation, $\odot$ represents the Hadamard product operation, and $\oplus$ represents the element-wise addition operation.

**Edge-aware Guidance Module (EGM).** EGM leverages the edge map $M_e$ as prior to guide the representation learning of object boundaries by mining the affinity between edge cue and region feature. As depicted in Fig. 5, the edge map $M_e \in \mathbb{R}^{1 \times H \times W}$ is first down-sampled by factor $d$ to the same size with the region feature $F_i \in \mathbb{R}^{C \times H_i \times W_i}$ ($H_i = \frac{H}{d}$ and $W_i = \frac{W}{d}$, $d \in \{1, 2, 4, 8\}$). Then, a $1 \times 1$ convolutional layer combined with a $3 \times 3$ depth-wise convolutional layer are performed on the down-sampled $M_e$ and $F_i$ ($i \in \{1, 2, 3, 4\}$) to obtain three features. These features are then reshaped to obtain $M_e^Q \in \mathbb{R}^{1 \times N_i}$, $F_i^K \in \mathbb{R}^{N_i \times C}$ and $F_i^V \in \mathbb{R}^{C \times N_i}$, respectively, where $N_i = H_i W_i$. Next, a matrix multiplication operation is performed between $M_e^Q$ and $F_i^K$, followed by a Softmax function, yielding the edge-aware affinity vector $E_a \in \mathbb{R}^{1 \times C}$:

$$
E_a = Softmax(M_e^Q \otimes F_i^K),
\tag{3}
$$

where $\otimes$ represents the matrix multiplication operation.

The element $E_a^l$ in the edge-aware affinity vector $E_a$ represents the channel correlation probability between the edge cue $M_e$ and the $l$-th channel in region feature $F_i$. Subsequently, a matrix multiplication operation between $E_a$ and $F_i^V$ is performed. The resulting feature is reshaped into $\mathbb{R}^{1 \times H_i \times W_i}$, and then combined with $F_i$ via a skip connection, thus resulting in the feature $F_o^i \in \mathbb{R}^{C \times H_i \times W_i}$. The computation process can be formulated as follows:

$$
F_o^i = \mathcal{R}(E_a \otimes F_i^V) \oplus F_i.
\tag{4}
$$

Finally, a channel attention layer and a $3 \times 3$ convolutional block are performed on $F_o^i$ to obtain the output feature $O_i \in \mathbb{R}^{C \times H_i \times W_i}$.

## 3.4 Loss Function

A combination of edge loss, region loss, boundary localization loss, and auxiliary loss is used to supervise our network.

**Edge Loss.** Considering the significantly larger number of negative samples than positive samples in edge maps, to overcome this imbalance, the dice loss [39] is used to supervise the EDB to obtain more discriminative edge feature. The edge loss is defined as:

$$
\mathcal{L}_{edge} = \frac{2 \sum_{x,y} \left(e_{x,y} \times E_{x,y}\right)}{\sum_{x,y} e_{x,y}^2 + \sum_{x,y} E_{x,y}^2},
\tag{5}
$$

where $e_{x,y}$ and $E_{x,y}$ represent the values of pixel $(x, y)$ in the coarse edge map $e$ and the edge ground truth $E$ obtained by an edge detector [23], respectively.

**Table 1: Quantitative results on four benchmarks. "F", "U", and "W" denote fully-supervised, unsupervised, and weakly-supervised methods, respectively. For the fully-supervised SOD methods shown in the top, as well as the fully-supervised COD methods shown in the middle, the best results are marked in bold. For the weakly-supervised and unsupervised methods shown in the bottom, the best and second best results are marked in Red and Blue, respectively.**

| Methods | Sup. | CAMO | | | | CHAMELEON | | | | COD10K | | | | NC4K | | | |
|---|---|---|---|---|---|---|---|---|---|---|---|---|---|---|---|---|---|
| | | $M \downarrow$ | $S_\alpha \uparrow$ | $E_\phi \uparrow$ | $F_\beta^\omega \uparrow$ | $M \downarrow$ | $S_\alpha \uparrow$ | $E_\phi \uparrow$ | $F_\beta^\omega \uparrow$ | $M \downarrow$ | $S_\alpha \uparrow$ | $E_\phi \uparrow$ | $F_\beta^\omega \uparrow$ | $M \downarrow$ | $S_\alpha \uparrow$ | $E_\phi \uparrow$ | $F_\beta^\omega \uparrow$ |
| PoolNet [21] CVPR'19 | F | 0.105 | 0.729 | 0.746 | 0.575 | 0.054 | 0.845 | 0.863 | 0.691 | 0.056 | 0.740 | 0.776 | 0.507 | 0.073 | 0.784 | 0.814 | 0.636 |
| EGNet [46] ICCV'19 | F | 0.109 | 0.732 | **0.800** | 0.604 | 0.065 | 0.797 | 0.860 | 0.649 | 0.061 | 0.736 | 0.810 | 0.517 | 0.075 | 0.777 | 0.841 | 0.639 |
| SCRN [38] ICCV'19 | F | **0.090** | **0.779** | 0.797 | **0.643** | 0.042 | 0.876 | 0.889 | 0.741 | 0.047 | **0.789** | 0.817 | 0.575 | 0.059 | **0.830** | 0.854 | 0.698 |
| F³Net [35] AAAI'20 | F | 0.109 | 0.711 | 0.741 | 0.564 | 0.047 | 0.848 | 0.894 | 0.744 | 0.051 | 0.739 | 0.795 | 0.544 | 0.070 | 0.780 | 0.824 | 0.656 |
| CSNet [9] ECCV'20 | F | 0.092 | 0.771 | 0.794 | 0.641 | 0.047 | 0.855 | 0.868 | 0.718 | 0.047 | 0.778 | 0.809 | 0.569 | 0.088 | 0.750 | 0.773 | 0.603 |
| UCNet [43] CVPR'20 | F | 0.094 | 0.739 | 0.787 | 0.640 | **0.036** | **0.880** | **0.930** | **0.817** | **0.042** | 0.776 | **0.857** | **0.633** | **0.055** | 0.811 | **0.871** | **0.729** |
| ITSD [48] CVPR'20 | F | 0.102 | 0.750 | 0.779 | 0.610 | 0.057 | 0.814 | 0.844 | 0.662 | 0.051 | 0.767 | 0.808 | 0.557 | 0.064 | 0.811 | 0.844 | 0.680 |
| SINet [5] CVPR'20 | F | 0.092 | 0.745 | 0.804 | 0.644 | 0.034 | 0.872 | 0.936 | 0.806 | 0.043 | 0.776 | 0.864 | 0.631 | 0.058 | 0.808 | 0.871 | 0.723 |
| UGTR [40] ICCV'21 | F | 0.086 | 0.784 | 0.822 | 0.684 | 0.031 | 0.888 | 0.910 | 0.794 | 0.036 | 0.817 | 0.852 | 0.666 | 0.052 | 0.839 | 0.874 | 0.746 |
| LSR [24] CVPR'21 | F | 0.080 | 0.787 | 0.838 | 0.696 | 0.030 | 0.890 | 0.935 | 0.822 | 0.037 | 0.804 | 0.880 | 0.673 | 0.048 | 0.840 | 0.895 | 0.766 |
| FAPNet [49] TIP'22 | F | 0.076 | 0.815 | 0.865 | 0.734 | 0.028 | 0.893 | 0.940 | 0.825 | 0.036 | 0.822 | 0.888 | 0.694 | 0.047 | 0.851 | 0.899 | 0.775 |
| BSANet [50] AAAI'22 | F | 0.079 | 0.794 | 0.851 | 0.717 | 0.027 | 0.895 | 0.946 | 0.841 | 0.034 | 0.818 | 0.891 | 0.699 | 0.048 | 0.841 | 0.897 | 0.771 |
| ZoomNet [29] CVPR'22 | F | 0.066 | 0.820 | 0.877 | 0.752 | 0.023 | 0.902 | 0.943 | 0.845 | 0.029 | 0.838 | 0.888 | 0.729 | 0.043 | 0.853 | 0.896 | 0.784 |
| DaCOD [34] MM'23 | F | **0.051** | **0.855** | **0.911** | 0.796 | - | - | - | - | **0.028** | 0.840 | 0.908 | 0.729 | **0.035** | 0.874 | 0.923 | 0.814 |
| FPNet [2] MM'23 | F | 0.056 | 0.852 | 0.905 | **0.806** | **0.022** | **0.914** | **0.961** | **0.856** | 0.029 | **0.850** | **0.913** | **0.748** | - | - | - | - |
| SAM [17] ICCV'23 | F | 0.132 | 0.684 | 0.687 | 0.606 | 0.081 | 0.727 | 0.734 | 0.639 | 0.050 | 0.783 | 0.798 | 0.701 | 0.078 | 0.767 | 0.776 | 0.696 |
| DUSD [45] CVPR'18 | U | 0.166 | 0.551 | 0.594 | 0.308 | 0.129 | 0.578 | 0.634 | 0.316 | 0.107 | 0.580 | 0.646 | 0.276 | - | - | - | - |
| USPS [27] NeurIPS'19 | U | 0.207 | 0.568 | 0.641 | 0.399 | 0.188 | 0.573 | 0.631 | 0.380 | 0.196 | 0.519 | 0.536 | 0.265 | - | - | - | - |
| SS [44] CVPR'20 | W | 0.120 | 0.673 | 0.762 | 0.545 | 0.065 | 0.772 | 0.858 | 0.662 | 0.065 | 0.678 | 0.764 | 0.469 | 0.087 | 0.718 | 0.800 | 0.587 |
| SCWS [41] AAAI'21 | W | 0.104 | 0.718 | 0.812 | 0.614 | 0.055 | 0.785 | 0.890 | 0.683 | 0.057 | 0.716 | 0.821 | 0.546 | 0.070 | 0.764 | 0.853 | 0.668 |
| CRNet [13] AAAI'23 | W | 0.092 | 0.735 | 0.815 | 0.641 | 0.046 | 0.818 | 0.897 | 0.744 | 0.049 | 0.733 | 0.832 | 0.576 | 0.063 | 0.775 | 0.855 | 0.688 |
| Ours | W | 0.091 | 0.750 | 0.840 | 0.669 | 0.044 | 0.825 | 0.910 | 0.749 | 0.049 | 0.749 | 0.840 | 0.596 | 0.061 | 0.793 | 0.869 | 0.709 |

**Region Loss.** The region loss ensures the predicted result align with the scribble region in the scribble ground truth. The region loss in this paper is partial cross-entropy loss, which is defined as:

$$\mathcal{L}_{reg} = \sum_{i \in S_r} -y_i \log \hat{y}_i - (1 - y_i) \log (1 - \hat{y}_i), \quad (6)$$

where $S_r$ is the scribble-labeled pixels set, $y$ represents the scribble ground truth, and $\hat{y}$ represents the predicted result, respectively.

**Boundary Localization Loss.** The boundary localization loss aims to guide the network to learn the localization of object boundaries. The boundary localization loss consists of the context affinity loss ($\mathcal{L}_{ca}$), the semantic significance loss ($\mathcal{L}_{ss}$), and the consistency loss ($\mathcal{L}_{cst}$). Detailed definitions of these three losses can be found in [13]. Overall, the boundary localization loss is defined as:

$$\mathcal{L}_{bl} = \mathcal{L}_{ca} + \mathcal{L}_{ss} + \mathcal{L}_{cst}. \quad (7)$$

**Auxiliary Loss.** To accelerate the network learning powerful feature representations, the auxiliary loss composed of the region loss and the context affinity loss is applied to the multi-level predicted results $M_2$, $M_3$, and $M_4$. The auxiliary loss is defined as:

$$\mathcal{L}_{aux} = \sum_{i=2}^{4} \alpha_i (\mathcal{L}_{reg}^i + \mathcal{L}_{ca}^i), \quad (8)$$

where $\alpha_i$ is a hyperparameter used to gradually decrease the weight of $M_i$ by setting its value to be $1 - 0.2(i - 1)$.

Finally, the total loss can be defined as:

$$\mathcal{L}_{total} = \mathcal{L}_{reg} + \mathcal{L}_{bl} + \mathcal{L}_{aux} + \beta \mathcal{L}_{edge}, \quad (9)$$

where $\beta$ is a hyperparameter used to trade off the edge loss with other losses. In our experiments, $\beta$ is set to 20.

## 4 Experiments

### 4.1 Experimental Setup

**Datasets.** Four popular benchmark datasets including CAMO [18], CHAMELEON [31], COD10K [5], and NC4K [24] are employed in our experiments. CAMO includes 1,250 samples in total, from which 1,000 samples are chosen for training and the rest 250 samples are used for testing. CHAMELEON contains 76 samples. COD10K contains 5,066 samples, which are divided into 3,040 training samples and 2,026 testing samples, respectively. NC4K is a large-scale COD testing dataset, comprising 4,121 samples. All samples in CHAMELEON and NC4K are used for testing only.

**Evaluation metrics.** We apply four widely-used metrics to evaluate the performance of different methods, including S-measure ($S_\alpha$) [3], mean E-measure ($E_\phi$) [4], weighted F-measure ($F_\beta^\omega$) [25], and Mean Absolute Error ($M$) [30]. Larger values of $S_\alpha$, $E_\phi$, and $F_\beta^\omega$ and smaller values of $M$ indicate better performance.

**Implementation details.** Our MiNet is implemented by using PyTorch framework and accelerated by NVIDIA A40 GPU. Due to the success of ResNet-50 [12] across multiple tasks [36, 37], we also use ResNet-50 pre-trained on ImageNet as the backbone, and the parameters of convolutional layers are initialized by kaiming-normal. Our model is optimized by SGD with momentum 0.9 and weight decay 0.0005. During the training phase, the model undergoes 180 epochs with a batch size of 10. We employ the triangle learning rate schedule with maximum learning rate 0.001. For both training and inference phases, the images are resized to $320 \times 320$. Horizontal flip is performed on input images to augment the training data.

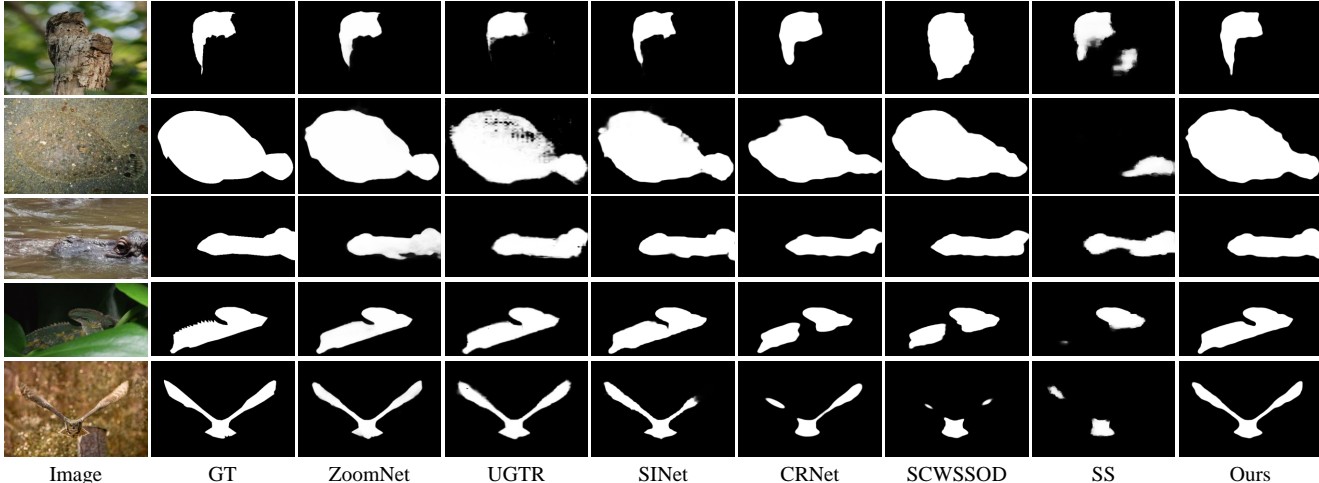

**Figure 6: Qualitative comparison of the proposed MiNet with three fully-supervised COD methods, one scribble-based weakly-supervised COD method, and another two scribble-based weakly-supervised SOD methods.**

## 4.2 Comparison with State-of-the-Art Methods

**Quantitative comparison.** Weakly-supervised camouflaged object detection (WSCOD) is an emerging research, so we compare our method with one scribble-based WSCOD method [13], two scribble-based weakly-supervised salient object detection (SOD) methods [41, 44] and two unsupervised SOD methods [27, 45]. We also report the results of 7 fully-supervised SOD methods [9, 21, 35, 38, 43, 46, 48] , 9 fully-supervised COD methods [2, 5, 17, 24, 29, 34, 40, 49, 50]. As shown in Table 1, our proposed MiNet consistently achieves the best performance compared with other weakly-supervised or unsupervised methods under all the evaluation metrics on four benchmarks. Compared with the recently presented method CRNet [13], our MiNet achieves the improvements of 2.2%, 1.9%, 1.8%, and 2.9 % in $M$, $S_\alpha$, $E_\phi$ , and $F_\beta^\omega$, respectively. In addition, the performance of our MiNet surpasses that of 5 fully-supervised SOD methods [9, 21, 35, 46, 48] and is comparable to some fully-supervised COD methods.

**Qualitative comparison.** Fig. 6 illustrates the qualitative comparisons between our proposed MiNet and several other methods, including three fully-supervised COD methods, i.e., ZoomNet [29], UGTR [40], and SINet [5], one scribble-based weakly-supervised COD method, i.e., CRNet [13], and two scribble-based weakly-supervised SOD methods, i.e., SCWS [41] and SS [44]. It is evident from the comparisons that our prediction results exhibit greater completeness of the camouflaged objects and achieve more accurate boundary localization compared to other scribble-based weakly-supervised methods. Furthermore, our proposed method is even comparable to the fully-supervised COD methods.

## 4.3 Ablation Study

In this section, we conduct some ablation studies on CAMO and COD10K to validate the effectiveness of our proposed MiNet.

**Impact of region and edge cues.** The mutual reinforcement between region and edge cues significantly enhances the accuracy of boundary predictions for camouflaged objects. To evaluate their

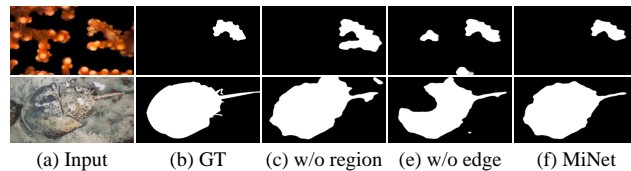

(a) Input    (b) GT    (c) w/o region    (e) w/o edge    (f) MiNet

**Figure 7: Visualizations showing region and edge cues effects.**

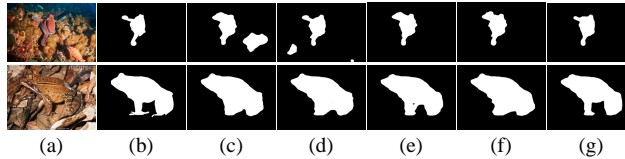

(a)    (b)    (c)    (d)    (e)    (f)    (g)

**Figure 8: Visual comparison of variant models incorporating different core modules. (a) Input; (b) GT; (c) Baseline (B); (d) B+EDB; (e) B+EDB+RGM; (f) B+EDB+EGM; (g) MiNet.**

individual impacts, we design two variants based on MiNet: 1) 'w/o region cue' (Table 2 ①), which removes the region feature $F_4$ and retains only the edge feature $F_e$ as input for the RGM; 2) 'w/o edge cue' (Table 2 ②), which eliminates the edge feature $F_e$ and employs only region features $F_i$ as input for the EGM at each layer. To maintain the structure of RGM and EGM, $F_e$ is used as the substitute of $F_4$ in RGM and $F_i$ is used as the substitute of $M_e$ in EGM. Furthermore, for the variant model 'w/o edge cue', the edge cue refinement net is removed.

As shown in Table 2, removing either region or edge cue leads to a deterioration in performance. Furthermore, visualizations in Fig. 7 also demonstrate the significance of both edge and region cues in enhancing the representation learning of object-related boundaries and mitigating the influence of non-object edge noise.

**Effectiveness of core modules.** We devise several variant models built upon the MiNet, so as to analyze the effectiveness of some

**Table 2: Ablation study on region and edge cues.**

| No. | Variants | CAMO | | | | COD10K | | | |
|---|---|---|---|---|---|---|---|---|---|
| | | $M \downarrow$ | $S_\alpha \uparrow$ | $E_\phi \uparrow$ | $F_\beta^\omega \uparrow$ | $M \downarrow$ | $S_\alpha \uparrow$ | $E_\phi \uparrow$ | $F_\beta^\omega \uparrow$ |
| ① | w/o region cue | 0.095 | 0.743 | 0.828 | 0.649 | 0.055 | 0.740 | 0.830 | 0.575 |
| ② | w/o edge cue | 0.099 | 0.738 | 0.822 | 0.645 | 0.056 | 0.733 | 0.827 | 0.564 |
| ③ | MiNet (Ours) | **0.091** | **0.750** | **0.840** | **0.669** | **0.049** | **0.749** | **0.840** | **0.596** |

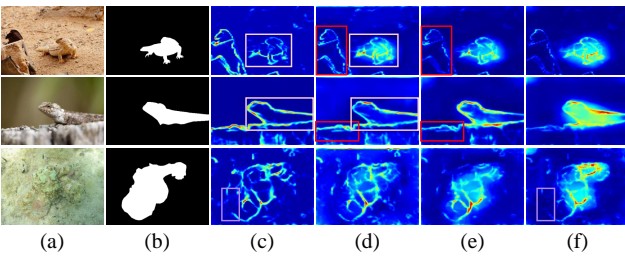

(a)  (b)  (c)  (d)  (e)  (f)

**Figure 9: Visualization of edge maps ($M_e$), which are progressively refined through PLM, PSS, and RCW operations. (a) Input; (b) GT; (c) Coarse edge feature; (d) $M_e$ with PLM; (e) $M_e$ with PLM and PSS; (f) $M_e$ with PLM, PSS, and RCW (MiNet).**

**Table 3: Ablation study on modules EDB, RGM, and EGM.**

| No. | EDB | RGM | EGM | CAMO | | | | COD10K | | | |
|---|---|---|---|---|---|---|---|---|---|---|---|
| | | | | $M \downarrow$ | $S_\alpha \uparrow$ | $E_\phi \uparrow$ | $F_\beta^\omega \uparrow$ | $M \downarrow$ | $S_\alpha \uparrow$ | $E_\phi \uparrow$ | $F_\beta^\omega \uparrow$ |
| ① | | | | 0.101 | 0.736 | 0.820 | 0.642 | 0.058 | 0.728 | 0.819 | 0.559 |
| ② | ✓ | | | 0.101 | 0.740 | 0.826 | 0.645 | 0.057 | 0.737 | 0.824 | 0.568 |
| ③ | ✓ | ✓ | | 0.095 | 0.744 | 0.825 | 0.658 | 0.051 | 0.745 | 0.837 | 0.592 |
| ④ | ✓ | | ✓ | 0.097 | 0.743 | 0.827 | 0.653 | 0.055 | 0.740 | 0.828 | 0.576 |
| ⑤ | ✓ | ✓ | ✓ | **0.091** | **0.750** | **0.840** | **0.669** | **0.049** | **0.749** | **0.840** | **0.596** |

core modules in the edge cue refinement net (contains the EDB and RGM) and region-boundary refinement net (contains the EGM). Specifically, the model after removing EDB, RGM, and EGM from the MiNet (Table 3 ⑤) serves as the baseline (Table 3 ①), and configurations of other variant models are presented in Table 3 ②-④. Compared with the baseline model (Table 3 ①), all three core modules bring certain gains. Particularly, compared to the model containing only EDB and using concatenation instead of EGM (Table 3 ②), the models incorporating the RGM to refine edge map (Table 3 ③) and the EGM to refine region features (Table 3 ④) on top of EDB bring more significant performance gains. Furthermore, when employing both RGM and EGM to refine edge and region cues (Table 3 ⑤), our final model achieves $F_\beta^\omega$ improvements of 4.2% and 6.6% on CAMO and COD10K, respectively, over the baseline model.

The visual comparisons of above models are presented in Fig. 8. We can observe that the coarse edge features introduced by EDB (Fig. 8(d)) lead to more accurate localization compared to the baseline model (Fig. 8(c)), but there are still some inaccurate boundary localization. Further enhancement is observed after refining the edge information with the RGM (Fig. 8(e)). Additionally, compared to simply embedding the edge information into the network using concatenation operation (Fig. 8(d)), using the EGM for edge information guidance yields better performance (Fig. 8(f)). Our final results (Fig. 8(g)) present the most similarity to GTs.

**Table 4: Ablation study on each operation in the RGM.**

| No. | PLM | PSS | RCW | CAMO | | | | COD10K | | | |
|---|---|---|---|---|---|---|---|---|---|---|---|
| | | | | $M \downarrow$ | $S_\alpha \uparrow$ | $E_\phi \uparrow$ | $F_\beta^\omega \uparrow$ | $M \downarrow$ | $S_\alpha \uparrow$ | $E_\phi \uparrow$ | $F_\beta^\omega \uparrow$ |
| ① | | | | 0.097 | 0.743 | 0.827 | 0.653 | 0.055 | 0.740 | 0.828 | 0.576 |
| ② | ✓ | | | 0.094 | 0.746 | 0.835 | 0.657 | 0.053 | 0.743 | 0.833 | 0.581 |
| ③ | ✓ | ✓ | | 0.095 | 0.746 | 0.828 | 0.659 | 0.052 | 0.745 | 0.836 | 0.588 |
| ④ | ✓ | ✓ | ✓ | **0.091** | **0.750** | **0.840** | **0.669** | **0.049** | **0.749** | **0.840** | **0.596** |

**Table 5: Ablation study on multi-level interaction manner.**

| No. | Interaction Manner | CAMO | | | | COD10K | | | |
|---|---|---|---|---|---|---|---|---|---|
| | | $M \downarrow$ | $S_\alpha \uparrow$ | $E_\phi \uparrow$ | $F_\beta^\omega \uparrow$ | $M \downarrow$ | $S_\alpha \uparrow$ | $E_\phi \uparrow$ | $F_\beta^\omega \uparrow$ |
| ① | only deepest-level | 0.097 | 0.735 | 0.833 | 0.646 | 0.056 | 0.727 | 0.827 | 0.558 |
| ② | only shallowest-level | 0.096 | 0.743 | 0.831 | 0.653 | 0.052 | 0.740 | 0.834 | 0.579 |
| ③ | multi-level | **0.091** | **0.750** | **0.840** | **0.669** | **0.049** | **0.749** | **0.840** | **0.596** |

**Impact of each operation in the RGM.** The PLM, PSS, and RCW operations within RGM play significant roles for acquiring discriminative edge map. Therefore, we conduct ablation studies to investigate their individual impacts. Results reported in Table 4 demonstrate the efficacy of each operation in the RGM.

Fig. 9 shows that the non-object noise can be effectively reduced through the combination of these operations. More specifically, based on the coarse edge feature (Fig. 9(c)), the PLM operation first highlights the object-related region (Fig. 9(d)). Then, the PSS operation suppresses the non-object noise within edge feature (Fig. 9(e)). Finally, the RCW operation further enhances the object-related region, thus obtaining the discriminative edge prior (Fig. 9(f)). The 3-*rd* row also illustrates an example with blurred boundaries, where highlighting the nearly absent boundary parts is still challenging.

**Efficacy of multi-level interaction.** We also investigate the efficacy of multi-level interaction within the region-boundary refinement net. Specifically, we devise two variant models utilizing single-level interaction: one that interacts edge map $M_e$ and region feature $F_4$ solely at the deepest level (Table 5 ①) and another that interacts edge map $M_e$ and region feature $F_1$ exclusively at the shallowest level (Table 5 ②). The performance of these two variant models, as depicted in Table 5, are both inferior to the model employing the multi-level interaction (Table 5 ③).

## 5 Conclusion

In this paper, we propose a novel Mutual Interaction Network (MiNet) for scribble-based weakly-supervised camouflaged object detection. The MiNet draws inspiration from human perception in discerning camouflaged objects and effectively exploits the mutual reinforcement between region and edge cues. To achieve this goal, we design the edge cue refinement net, which includes the EDB for generating the coarse edge feature and the RGM for highlighting object-related areas within the edge feature, under the guidance of region cue. Leveraging the enhanced edge cue, we further devise the region-boundary refinement net to refine the object boundaries in an iterative and multi-level manner. Specifically, the EGM is developed and integrated as a core module in this sub-network to incorporate edge cue with each level of region feature. Experiments on four benchmarks demonstrate that our proposed MiNet outperforms the state-of-the-art methods.

## Acknowledgments

This work was supported in part by the National Natural Science Foundation of China under Grants 62072110, 61972097, 62171134, and U21A20472, in part by the Natural Science Foundation of Fujian Province, China under Grants 2023J01067 and 2020J01494, in part by the Major Science and Echnology Project of Fujian Province (China) under Grant 2021HZ022007, and in part by Industry-Academy Co-operation Project under Grant 2021H6022.

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
