# OpenReview forum: "MiNet: Weakly-Supervised Camouflaged Object Detection through Mutual Interaction between Region and Edge Cues"
_acmmm.org/ACMMM/2024/Conference — MM2024 Poster_

### Official Review · Reviewer_t3et · 2024-05-14

**Rating:** 6
**Confidence:** 4

**Summary:**

This paper starts from the observation that existing scribble-based camouflage object detection methods suffer from inaccurate object boundary detection due to insufficient and imprecise boundary supervision in scribble annotations. To address this, we propose a novel Mutual Interaction Network (MiNet) that fosters mutual reinforcement between region and edge cues, thus incorporating stronger prior knowledge to boost detection accuracy.

**Strengths:**

This paper's scribble-based weakly supervised method effectively addresses the issue of insufficient manual annotations. Meanwhile, the region-aware guidance module in the refinement network enhances the discriminative power of edges by utilizing object boundaries. Finally, extensive experiments validate the reliability of the proposed method.

**Limitations:**

The overall idea of this paper is clear, aiming to improve accuracy by prioritizing the determination of object boundaries. However, in the detection of hidden objects, if the image is of low clarity or the object boundaries are blurred, does the method of prioritizing boundary determination still maintain reliability?

It is well known that weakly supervised methods are sensitive to noise and increase the consumption of computational resources. Are there any relevant explanations or experiments to verify the explanation for the above-mentioned issues?

**Suitability:**

3

---

### Official Review · Reviewer_waEg · 2024-05-14

**Rating:** 4
**Confidence:** 3

**Summary:**

This paper proposes a Mutual Interaction Network (MiNet) for scribble-based weakly-supervised camouflaged object detection (WSCOD). The core region-aware guidance module (RGM) and core edge-aware guidance module (EGM) are constructed, where the former is used to highlight object-relevant regions in edge features, and the latter employs discriminative edge maps as guidance to focus on object boundaries in a finer manner, capturing more accurate object boundaries through iterative refinement. Experimental results on four benchmark datasets demonstrate that the proposed method outperforms existing state-of-the-art approaches.

**Strengths:**

- The experiments are substantial, and a wealth of visualizations intuitively demonstrate the superiority of the proposed method.
- The content of the article is logically concise, well-structured, and easy to comprehend.

**Limitations:**

- In Section 3.4, it is questioned whether there is an overlap between $\mathcal{L}\_{ca}$ in Equation 7 and $\mathcal{L}\_{ca}^{i}$ in Equation 8, and how they are related and differentiated. It is suggested to clarify this aspect.
- In Section 4.2, Table 1, it is suggested to add the results of FPNet [1] and DaCOD [2] in the table.
- In Section 4.3, Figure 8, the order of e and f is reversed in the figure.
- In Section 4.3, Table 2, it is recommended to add the results of the model in settings where neither region cue nor edge cue is present.


[1] Frequency Perception Network for Camouflaged Object Detection, ACM MM, 2023.

[2] Depth-aided Camouflaged Object Detection, ACM MM, 2023.

**Suitability:**

2

---

### Official Review · Reviewer_rJNz · 2024-05-24

**Rating:** 4
**Confidence:** 4

**Summary:**

This paper proposes a novel Mutual Interaction Network (MiNet) for scribble-based weakly-supervised camouflaged object
detection. The MiNet draws inspiration from human perception in discerning camouflaged objects and effectively exploits the mutual
reinforcement between region and edge cues.

**Strengths:**

This paper proposes a novel Mutual Interaction Network (MiNet) for scribble-based weakly-supervised camouflaged object
detection.  Overall it's interesting and novel, and the writing is pretty good.

**Limitations:**

Although this work is well done, I still have the following questions that the author needs to answer:
1) Why do edge features need to integrate the four features from the backbone network? As far as I know, existing work has confirmed that using the feature integration of the first and last layers to learn edge features is very effective and can reduce noise features. , please refer to the BGNet model of IJCAI2022. Will your approach add non-edge noise features from the input end? If so, the following steps will be redundant. Please add experiments to prove the effectiveness of your input. In addition, why the edge ground truth map is generated using the edge detector in [22]? From Figure 2, the edge map effect is not very good and the edges are rough. Will this not affect the final performance? as far as I know. A lot of work is done to generate edge truth maps from ground truth maps of camouflaged objects, please explain;
2) There are some differences between the architecture diagram of Edge Cue Refinement Net in Figure 3 and the detailed diagram in Figure 4. For example, is the Probability Map Calculation in Figure 3 missing an input arrow Fr1? There are also some symbols whose full names are not explained, such as Does the CALayer in Figure 4 refer to the channel attention layer?
3) I don’t quite understand Si in line 454. It refers to taking a fixed range of pixel sets, then selecting the pixel with the highest pixel value in this range, and then assigning a weight of 1? Why only one pixel is allocated? Please explain?
4) In the ablation experiment part, there is no ablation experiment on Boundary Localization Loss? I have reason to suspect that the performance improvement comes from them. Please add this part of the ablation experiment? This is the part I am most concerned about. If this part is removed and the experimental results are still good, it can prove the effectiveness of your method. After all, your method is slightly complicated.
5) Please add a performance comparison with the SAM model. I would like to see the results compared with this large visual model.

If the above questions are explained, I will increase my score.

**Suitability:**

2

---

### Official Review · Reviewer_sed8 · 2024-05-26

**Rating:** 4
**Confidence:** 3

**Summary:**

The manuscript enhances the performance of weakly supervised COD tasks by aggregating edge information, innovatively employing the strategy of aligning edge features with region features to suppress non-object edges. Experimental results achieve state-of-the-art performance on all datasets, and visualizations further demonstrate the effectiveness of the proposed method.

**Strengths:**

This paper is well written and easy to follow.
The technical contributions are clarified clearly and it should be possible to reproduce the method.
Extensive experiments show the state-of-the-art performance achieved by the proposed method.

**Limitations:**

1. Comprehending the PLM operation depicted in Figure 4 poses a challenge. How might one reconcile the matching of a region's feature map with that of an edge to delineate a highlighted area, as evidenced by the yellow anchor box position in Figure 3?

2. In the PLM operation, might it be advisable to opt for the F1 feature for processing? Considering the network structure, it appears that the F1 feature should offer finer granularity.

3. Figure 9 showcases successful instances of the RGM module. It may be advisable to also present some unsuccessful cases and elucidate the reasons for their lack of success.

4. In comparison to other weakly supervised methods, the manuscript's experimental results demonstrate improvement. Could this be attributed to its increased utilization of edge information compared to other methods? The second row in Table 2 presents the experimental results without using edge cues. Can this result be considered a fair comparison with other methods, indicating the performance without utilizing edge information?

**Suitability:**

2

---

### Meta-Review · Area_Chair_QSXg · 2024-06-24

**Recommendation:** Accept (Poster)
**Confidence:** 5

**Metareview:**

This submission initially received four positive reviews, leading to the decision to accept the submission.